# Efficient Perceiving Local Details via Adaptive Spatial-Frequency Information Integration for Multi-focus Image Fusion

## ABSTRACT

Multi-focus image fusion (MFIF) aims to combine multiple images with different focused regions into a single all-in-focus image. Existing unsupervised deep learning-based methods only fuse structural information of images in the spatial domain, neglecting potential solutions from the frequency domain exploration. In this paper, we make the first attempt to integrate spatial-frequency information to achieve high-quality MFIF. We propose a novel unsupervised spatial-frequency interaction MFIF network named SFIMFN, which consists of three key components: Adaptive Frequency Domain Information Interaction Module (AFIM), Ret-Attention-Based Spatial Information Extraction Module (RASEM), and Invertible Dual-domain Feature Fusion Module (IDFM). Specifically, in AFIM, we interactively explore global contextual information by combining the amplitude and phase information of multiple images separately. In RASEM, we design a customized transformer to encourage the network to capture important local high-frequency information by redesigning the self-attention mechanism with a bidirectional, two-dimensional form of explicit decay. Finally, we employ IDFM to fuse spatial-frequency information without information loss to generate the desired all-in-focus image. Extensive experiments on different datasets demonstrate that our method significantly outperforms state-of-the-art unsupervised methods in terms of qualitative and quantitative metrics as well as the generalization ability.

## CCS CONCEPTS

• **Computing methodologies** → **Image representations**.

## KEYWORDS

multi-focus, image fusion, spatial-frequency interaction, customized transformer

## 1 INTRODUCTION

Constrained by the focused capability of optical imaging devices, objects may appear blurred in local regions due to being out of the depth-of-field (DoF) during the imaging process. To this end, the multi-focus image fusion (MFIF) aims to extract complementary information from images with multiple focused regions to generate an all-in-focus image. MFIF has been applied to many applications such as microscopic imaging [21, 31], image segmentation [55], image classification [9] and image recognition [16].

**Unpublished working draft. Not for distribution.**

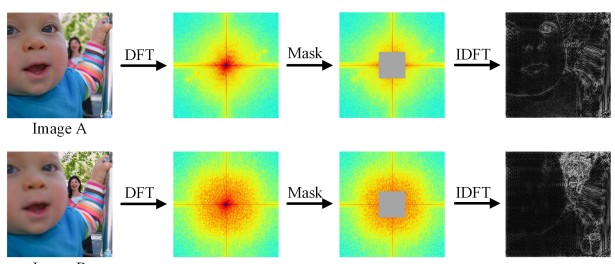

**Figure 1: After transforming images into the frequency domain using discrete Fourier transform (DFT), the edges of focused regions can be highlighted by filtering signals at appropriate frequencies (the child in source image A and the woman in source image B).**

Traditional MFIF algorithms are typically categorized into spatial domain-based methods and transform domain-based methods. In the former method, an all-in-focus image is obtained by weighting the content of the source images [4, 19, 22, 37, 52]. These methods typically have lower computational complexity, but their performance heavily depends on hand-made prior. The transform domain-based methods first convert the source images into the transform domain, then fuse the transformed coefficients, and finally obtain the fused image through the corresponding inverse transformation. The typical methods include sparse representation methods [44, 50], multi-scale methods [2, 5, 17], gradient domain-based methods [30, 53] and hybrid methods [24]. However, after undergoing domain transformation, coefficient fusion, and inverse transformation, the attenuation of the signal and the accumulation of errors become particularly evident. Moreover, most traditional methods often fail to fully consider the local gradient changes in the source images, leading to challenges such as correctly identifying small defocused (focused) regions within larger focused (defocused) areas.

In recent years, many deep learning-based MFIF methods [8, 27, 28, 47, 49] have emerged. These methods employ deep networks to learn priors from numerous training samples. However, it is challenging to collect all-in-focus image data in practical scenarios, making it challenging to train deep models in a supervised manner. Existing deep learning-based methods operate on the source images in the spatial domain. However, by applying the discrete Fourier transform (DFT) to convert images into the frequency domain, we observe that the edges of focused regions can be highlighted by filtering signals at appropriate frequencies as shown in Figure 1. Motivated by this observation, we aim to investigate potential unsupervised MFIF approaches in the frequency domain.

Different from the local receptive field property of convolutional operator, the visual transformer (ViT) capture long-range dependencies by employing the multi-head global attention mechanism among different ordered input feature segments [7, 34, 35, 38, 54].

In this paper, we aim to leverage ViT to establish long-range dependencies between multiple images focusing on different regions. Recently, Retentive Network (RetNet) [33] has garnered significant attention in the field of natural language processing (NLP), primarily due to its explicit decay mechanism. In MFIF, our objective is to enable the network to accurately detect the critical focused areas in each source image. To this end, we attempt to redesign RetNet into a 2D form and integrate it with ViT to make it applicable to image data.

Based on the above analysis, we first attempt to investigate the MFIF task from the perspective of spatial-frequency information integration. We design a novel unsupervised MFIF network that efficiently perceives the local details of different source images through the interaction of spatial and frequency domains. It comprises three core components: Adaptive Frequency Domain Information Interaction Module (AFIM), Ret-Attention-Based Spatial Information Extraction Module (RASEM), and Invertible Dual-domain Feature Fusion Module (IDFM). Specifically, in AFIM, after transforming paired source images into the frequency domain through DFT, we separate their amplitude and phase information. These components are further interactively integrated to explore the global contextual details of the fused images. In RASEM, we design a customized transformer with bidirectional, 2D explicit decay self-attention mechanism, used to capture long-range dependencies among features of multiple source images while effectively perceiving local focused regions in each image. Finally, an invertible neural network information fusion module IDFM is introduced to avoid information loss during the spatial-frequency domain features fusion process. Extensive experiments on different datasets demonstrate that our proposed method significantly outperforms state-of-the-art unsupervised methods in terms of quantitative metrics, visual quality, and generalization ability. Our contributions can be summarized as follows:

- We propose a novel unsupervised MFIF framework SFIMFN that adaptively integrates high-low frequency information from the spatial and frequency domains of multiple source images. To the best of our knowledge, this is the first attempt to investigate the MFIF task from the perspective of spatial-frequency information integration.
- We design a customized transformer for MFIF. By redesigning the self-attention mechanism into a bidirectional, two-dimensional form of explicit decay, the network can perceive the locally focused regions more effectively.
- Extensive experiments on different datasets demonstrate that our method significantly outperforms SOTA unsupervised methods. The necessity and effectiveness of each module also be further demonstrated through ablation experiments.

## 2 RELATED WORK

### 2.1 Spatial domain-based MFIF methods

Spatial domain-based methods primarily rely on the focus measure, which compute directly in the spatial domain to generate the fused image based on the decision map with high efficiency. The spatial domain-based methods can be further divided into pixel-based [3, 25], block-based [10], and region-based methods [19]. Pixel-based and block-based approaches rely on pixel activity-level

measurement function or algorithms to evaluate the pixel activity and obtain a rule-based saliency map, generating a focus decision map for each source image [29, 32]. De et al. [10] first introduced the quadtree decomposition into the MFIF. Later, Wang et al. [36] proposed an MFIF model based on quad-tree decomposition and edge-weighted focus detection to decompose the source image into appropriately sized blocks. In contrast, region-based algorithms can provide more precise differentiation between the focused regions and the defocused regions, but the fusion performance heavily relies on the segmentation algorithm [11, 12]. Accurately determining pixel focus capability can be challenging when artifacts and boundary effects are incorrectly assessed, possibly leading to suboptimal visual results.

### 2.2 Transform domain-based MFIF methods

Transform domain-based methods mainly consist of three stages: decomposing the source image into a series of multi-scale high-low frequency coefficients, designing different fusion rules for coefficient fusion, and finally reconstructing the selected coefficients to obtain the fusion results. Since Burt et al. [6] first proposed a MFIF method based on the Laplacian pyramid, various multi-scale decomposition methods have been used for image fusion, including the discrete wavelet transform (DWT) [43], discrete cosine transform (DCT) [2] and non-subsampled contourlet transform (NSCT) [1]. In summary, transform domain-based methods perform well in preserving edge details and boundaries due to their similarity to human visual processing, but their sensitivity to high-frequency components can lead to image distortion if not handled carefully.

### 2.3 Deep learning-based MFIF methods

Benefiting from the powerful representation capability of deep neural networks, some deep learning-based MFIF methods have been proposed, which can be categorized into supervised [13, 20, 23, 45, 46] and unsupervised methods [8, 15, 28, 39–41, 48]. In terms of supervised models, Liu et al. [23] proposed a classification-based image fusion model, introducing the convolutional neural network (CNN) into MFIF firstly. Recently, Li et al. [20] presented a diffusion-based MFIF method named FusionDiff, which used diffusion model to fuse two source images by iteratively performing multiple denoising operations. However, real multi-focus image datasets are severely lacking. Xu et al. introduced a new unsupervised model for MFIF based on gradients and connected regions [39]. Then they designed a unified densely connected network [41] for different types of image fusion tasks. Zhang et al. [48] proposed a new unsupervised GAN-based model with adaptive and gradient joint constraints for MFIF by extracting and reconstructing information. Hu et al. [15] proposed a novel framework ZMFF that use parameterized networks to successfully mine the deep priors of clear fused image and the corresponding focus maps. Ma et al. [28] introduced a method based on CNN and SwinTransformer [26] to extract features containing both local and global information and fuse these features intra-domain and cross-domain. However, the above methods all operate on images in the spatial domain, neglecting to explore contextual information from the frequency domain.

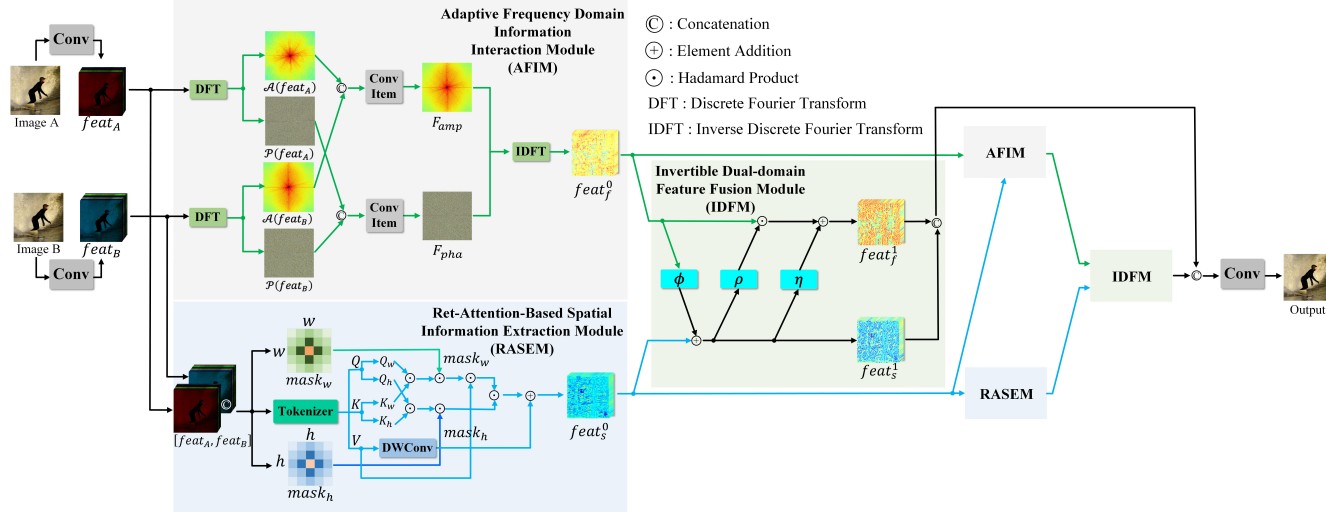

**Figure 2: The overall framework of SFIMFN. It consists of three key parts: Adaptive Frequency Domain Information Interaction Module (AFIM), Ret-Attention-Based Spatial Information Extraction Module (RASEM), and Invertible Dual-domain Feature Fusion Module (IDFM).**

## 3  METHOD

Figure 2 shows the overall architecture of our spatial-frequency interaction MFIF network SFIMFN, which mainly consists of three parts, Adaptive Frequency Domain Information Interaction Module (AFIM), Ret-Attention-Based Spatial Information Extraction Module (RASEM), and Invertible Dual-domain Feature Fusion Module (IDFM). The details will be illustrated below.

### 3.1  Adaptive Frequency Domain Information Interaction Module

Fourier transform is commonly utilized to analyze the frequency components of images. When dealing with images that have multiple color channels, the Fourier transform is computed independently for each color channel. Given an image $x \in R^{H \times W \times C}$, the Fourier transform $\mathcal{F}$ transfers it to Fourier domain as the complex component $\mathcal{F}(x)$:

$$\mathcal{F}(x)(u,v) = \frac{1}{\sqrt{HW}} \sum_{h=0}^{H-1} \sum_{w=0}^{W-1} x(h,w) e^{-j2\pi \left( \frac{h}{H}u + \frac{w}{W}v \right)}, \quad (1)$$

The amplitude component $\mathcal{A}(x)(u,v)$ and the phase component $\mathcal{P}(x)(u,v)$ are expressed as:

$$\mathcal{A}(x)(u,v) = \sqrt{R^2(x)(u,v) + I^2(x)(u,v)}, \quad (2)$$

$$\mathcal{P}(x)(u,v) = \arctan \left[ \frac{I(x)(u,v)}{R(x)(u,v)} \right], \quad (3)$$

where $R(x)$ and $I(x)$ represent the real and imaginary part of $\mathcal{F}(x)$ respectively. In this paper, the Fourier transform and its inverse process are independently computed on each channel of the feature maps. In AFIM, for two source images $A$ and $B$, which are focused on different areas, we first conduct shallow feature extraction on

each of them using $1 \times 1$ convolutional layers:

$$\begin{aligned} feat_A &= Conv_{1\times1}(A), \\ feat_B &= Conv_{1\times1}(B), \end{aligned} \quad (4)$$

then we get their amplitude and phase information individually through discrete Fourier transform (DFT):

$$\begin{aligned} \mathcal{A}(feat_A), \mathcal{P}(feat_A) &= \mathcal{F}(feat_A), \\ \mathcal{A}(feat_B), \mathcal{P}(feat_B) &= \mathcal{F}(feat_B), \end{aligned} \quad (5)$$

where $\mathcal{A}(\cdot)$ and $\mathcal{P}(\cdot)$ indicate the amplitude and phase respectively. Then, we integrate the amplitude and phase information of the two images separately, and use two convolutional networks to learn the fused amplitude and phase features, respectively:

$$\begin{aligned} F_{amp} &= CN(Cat(\mathcal{A}(feat_A), \mathcal{A}(feat_B))), \\ F_{pha} &= CN(Cat(\mathcal{P}(feat_A), \mathcal{P}(feat_B))), \end{aligned} \quad (6)$$

where $F_{amp}$ and $F_{pha}$ are the fused amplitude and phase features, respectively. $CN(\cdot)$ represents a simple convolutional network. The interaction of frequency domain components enhances the global frequency representation. Subsequently, we employ the inverse discrete Fourier transform (IDFT) to convert the fused amplitude and phase components of $F_{amp}$ and $F_{pha}$ back to the spatial domain:

$$feat_f = \mathcal{F}^{-1}\left( F_{amp}, F_{pha} \right). \quad (7)$$

where $\mathcal{F}^{-1}(\cdot)$ is the IDFT operation and $feat_f$ represents the global information representation obtained after information processing in the Fourier domain.

### 3.2  Ret-Attention-Based Spatial Information Extraction Module

We aim to utilize transformer to establish long-range dependencies among multiple source images, enhancing the boundaries between different focused regions. In MFIF, the high-frequency signals in

the focused regions are significantly more pronounced than in the unfocused regions. Therefore, we attempt to extend the unidirectional, explicit decay self-attention mechanism from RetNet to a bidirectional, two-dimensional form, making neighboring pixels in the focused local regions to provide more information to each other. In RetNet, the retention layer is defined as:

$$Q = (XW_Q) \odot \Theta, \quad K = (XW_K) \odot \overline{\Theta}, \quad V = XW_V$$

$$\Theta_n = e^{in\theta}, \quad D_{nm} = \begin{cases} \gamma^{n-m}, & n \geq m \\ 0, & n < m \end{cases} \tag{8}$$

$$\text{Retention}(X) = (QK^{\mathsf{T}} \odot D)V,$$

where $\gamma, \theta \in \mathbb{R}^d$ are both scalar, $n$ and $m$ represent the indices of tokens, $\overline{\Theta}$ is the complex conjugate of $\Theta$, and $D \in \mathbb{R}^{|x| \times |x|}$ combines causal masking and exponential decay along relative distance as one matrix. It also can be written as:

$$o_n = \sum_{m=1}^{n} \gamma^{n-m} \left(Q_n e^{in\theta}\right) \left(K_m e^{im\theta}\right)^{\dagger} v_m, \tag{9}$$

To adapt the retention for image data, we first extend the retention to two dimensions, where for each token, its output becomes:

$$o_n = \sum_{m=1}^{N} \gamma^{|n-m|} \left(Q_n e^{in\theta}\right) \left(K_m e^{im\theta}\right)^{\dagger} v_m, \tag{10}$$

where $N$ is the number of tokens. It also can be written as:

$$BiRet(X) = \left(QK^{\mathsf{T}} \odot D^{Bi}\right)V,$$
$$D_{nm}^{Bi} = \gamma^{|n-m|}, \tag{11}$$

where $BiRet(\cdot)$ denotes the retention with bidirectional modeling ability. We further extend the one-dimensional retention to two dimensions. We represent the two-dimensional coordinate of the

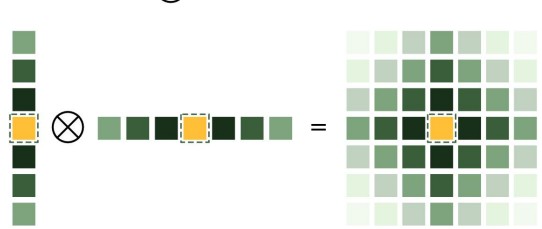

**Figure 3: Illustration of $D^{2d}$.**

$n$-th token as $(x_n, y_n)$. As shown in Figure 3, based on the 2D coordinates of each token, we modify each element in the matrix $D$ to be the Manhattan distance between the corresponding token pairs at their respective positions. Thus, the 1D decay coefficients can be transformed into 2D form:

$$D_{nm}^{2d} = \gamma^{|x_n - x_m| + |y_n - y_m|}, \tag{12}$$

For the joint embedding $X(A, B)$ of the source images $A$ and $B$, we generate its corresponding joint queries $Q_{AB}$, keys $K_{AB}$, and values $V_{AB}$. Finally, we use Softmax to introduce nonlinearity to the network to get the spatial feature $feat_s$:

$$feat_s = \left(Softmax\left(Q_{AB}K_{AB}^{\mathsf{T}}\right) \odot D^{2d}\right)V_{AB}. \tag{13}$$

In RASEM, a customized transformer is designed to establish long-range dependencies between multiple source images and enhance the capacity to perceive local high-frequency signals.

## 3.3 Invertible Dual-domain Feature Fusion Module

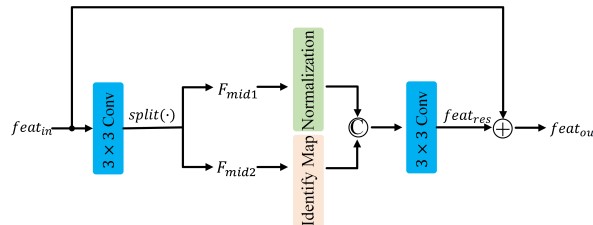

**Figure 4: The details of $\rho(\cdot)$ and $\eta(\cdot)$.**

Different from pure convolution layers, the invertible network have the property of information-lossless during the information transformation process. In IDFM, we aim to avoid information loss during the fusion of frequency domain feature $feat_f$ and spatial domain feature $feat_s$. As detailed in the Figure 2, given spatial feature $feat_s^0$ and frequency domain feature $feat_f^0$, the output of IDFM will be calculated as:

$$feat_f^1 = feat_f^0 \odot exp\left(\rho\left(feat_s^1\right)\right) + \eta\left(feat_s^1\right), \tag{14}$$

$$feat_s^1 = feat_s^0 + \phi\left(feat_f^0\right), \tag{15}$$

where $exp(\cdot)$ is Exponential function in mathematical, and $\rho(\cdot)$ and $\eta(\cdot)$ represent the scale and translation functions from the channels of frequency domain feature $feat_f^0$ to the channels of spatial feature $feat_s^0$, respectively. $\odot$ is the Hadamard product. Note that functions $\rho(\cdot)$ and $\eta(\cdot)$ are not necessarily invertible, so we implement them through neural networks. As shown in Figure 4:

$$feat_{mid} = Conv_{3\times3}(feat_{in}), \tag{16}$$

$$feat_{mid1}, feat_{mid2} = split(feat_{mid}), \tag{17}$$

$$feat_{res} = Conv_{3\times3}((Norm(feat_{mid1}), feat_{mid2})), \tag{18}$$

$$feat_{out} = feat_{res} + feat_{in}. \tag{19}$$

we first use a $3 \times 3$ convolution to project input features $feat_{in}$ to intermediate features $feat_{mid}$, then $feat_{mid}$ are divided into two parts. The first part $feat_{mid1}$ is normalized by Normalization operation and then concatenates with $feat_{mid2}$ in channel dimension. Next, after a $3 \times 3$ convolution the features $feat_{res}$ are obtained. Finally, the invertible block output the enhanced feature $feat_{out}$ by adding $feat_{res}$ with shortcut features $feat_{in}$. In this paper, we cascade two dual-domain information extraction-fusion modules and finally use a $1 \times 1$ convolution layer to generate the final all-in-focus image.

 

## 3.4 Loss Functions

In this paper, we consider the similarity between the fused image and the source images in terms of pixel density, gradient information, and structure together. Three loss terms are formulated as:

$$\mathcal{L}_{pix} = \|Y - A\|_F^2 + \|Y - B\|_F^2, \tag{20}$$

$$\mathcal{L}_{grad} = \|\nabla Y - \nabla A\|_F^2 + \|\nabla Y - \nabla B\|_F^2, \tag{21}$$

$$\mathcal{L}_{ssim} = 2 - SSIM(Y, A) - SSIM(Y, B), \tag{22}$$

where $\|\cdot\|_F$ denotes the Frobenius norm, $\nabla$ is the gradient operator. The total loss is formulated as:

$$\mathcal{L}_{total} = \lambda_1 \mathcal{L}_{pix} + \lambda_2 \mathcal{L}_{grad} + \lambda_3 \mathcal{L}_{ssim}. \tag{23}$$

where $\lambda_1$, $\lambda_2$, and $\lambda_3$ are weight factors.

## 4 EXPERIMENTS

### 4.1 Baseline Methods

We compared the performance of our method with both traditional MFIF methods and deep learning-based MFIF methods. We selected two traditional MFIF methods, including SFMD [18] and DCT_Corr [2]. The deep learning-based methods consist of three supervised methods: MGDN [13], MFFT [46], FusionDiff [20], and three unsupervised methods: SwinFusion [28], ZMFF [15] and MUFusion [8].

### 4.2 Implementation Details

We implemented our network on the PC with a single NVIDIA GeForce RTX 3090, and we built our network in Pytorch framework. The parameters of our network are updated by the Adam optimizer. The learning rate, batch size and the epoch are set to $1 \times 10^{-4}$, 20 and 10 respectively.

### 4.3 Dataset and Evaluation Metrics

**Dataset.** Our experiments are conducted on three datasets, the MFI-WHU dataset [48], the Lytro dataset [29] and the MFFW dataset [42]. The MFI-WHU dataset is obtained by synthesis, which contains 120 near-focused and far-focused image pairs and full-clear images. While the Lytro dataset is created based on light field data, containing 20 near-focused and far-focused image pairs together with 4 sequences of different scenes. The MFFW dataset includes 13 real multi-focus image pairs with strong defocus spread effect (DSE). To ensure the fairness of the experiments, both supervised and unsupervised methods are trained on the MFI-WHU dataset, which provides ground-truths and then test on the Lytro and the MFFW respectively. We conduct ablation experiments on the Lytro dataset. We crop the images into $128 \times 128$ patches for training, while the entire image is used as input for testing.

**Metrics.** We use 10 widely-used image quality assessment (IQA) metrics to evaluate the fusion performance of MFIF, namely entropy (EN), mutual information entropy (MI), spatial frequency (SF), average gradient (AG), standard deviation (SD), correlation coefficient (CC), visual information fidelity (VIF), edge based fidelity ($Q_{abf}$), peak signal-to-noise ratio (PSNR) and structural similarity index measure (SSIM) [51].

## 4.4 Comparison with SOTA Methods

We compared our method with baseline methods in terms of quantitative metrics and visual quality. As shown in Fig. 5, for the first pair of images from the Lytro dataset, upon zooming in on local regions of the fused image, varying degrees of artifacts can be observed in the fusion results by competing methods. In contrast, our method produces sharper boundaries (such as the hat brim of the monkey). Similarly, for the second pair of images, our method makes it easier to distinguish the boundary between the lighthouse and the sky. This is because our approach leverages frequency-domain information to enhance the interaction of global contextual information, while RASEM weakens the influence between unrelated objects. Similar to what is shown in Fig. 6, in the first pair of images from the MFFW dataset, our method is capable of preserving the details of the local grass (foreground) while retaining the texture on the wooden planks (background). In comparison, the grass generated by ZMFF and DCT_Corr both exhibit color distortion, while FusionDiff, SwinFusion, and MUFusion fail to preserve high-quality fine-grained details of the grass. This further underscores the advantage of our method in preserving both global and local textures.

We also calculated the average values of ten IQA metrics for these methods, for quantitative comparison. As shown in Table 1 and Table 2, our method outperforms other SOTA unsupervised MFIF methods. VIF measures the information fidelity of the fused image, which is consistent with the human visual system [14]. The performance on the VIF metric demonstrate that our method preserves the pixel density of different focused regions with the highest quality, surpassing the second place by 0.013 and 0.012 on two datasets, respectively. The EN and MI metrics show the superiority of our method in this fusion task from the point of information amount and correlations with source images, respectively. The performance on these two metrics indicates that our fusion results can preserve the information of the source images to the maximum extent.

To ensure fairness in the experiments, all methods were trained on the MFI-WHU dataset, which includes ground truths, and then tested on the other two datasets. The metrics in Table 1 and Table 2 demonstrate that the generalization ability of our method significantly outperforms existing unsupervised methods.

## 4.5 Ablation Experiments

Adaptive Frequency Domain Information Interaction Module (AFIM), Ret-Attention-Based Spatial Information Extraction Module (RASEM), and Invertible Dual-domain Feature Fusion Module (IDFM) are three key modules of SFIMFN, we conducted a series of ablation experiments on the Lytro dataset to demonstrate their effectiveness and necessity. Additionally, we also conducted ablation experiments to verify the effectiveness of the 2D Ret-Attention mechanism and three loss terms proposed in this paper.

**Adaptive Frequency Domain Information Interaction Module.** AFIM is utilized to explore the edge differences between different focused regions. To demonstrate the effectiveness of AFIM, we replaced AFIM with RASEM while keeping the network parameters at the same level. Table 3 shows that replacing AFIM with RASEM results in a decrease in all IQA metrics, especially SF and $Q_{abf}$. This is because AFIM influences the global structural information of fusion results in the frequency domain. The lack of global context

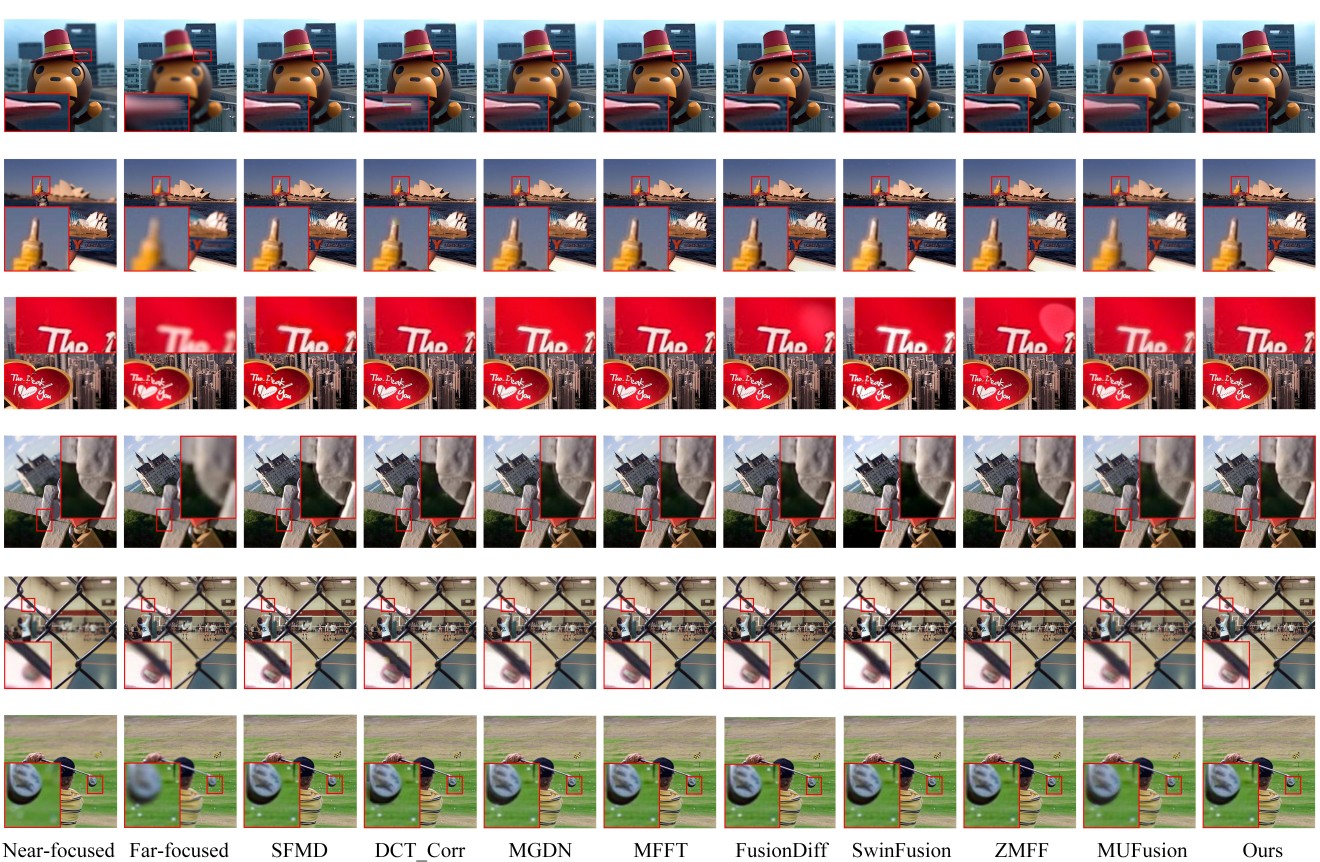

Near-focused   Far-focused   SFMD   DCT_Corr   MGDN   MFFT   FusionDiff   SwinFusion   ZMFF   MUFusion   Ours

**Figure 5: The visual comparisons between other MFIF methods and our method on the Lytro dataset.**

**Table 1: The average scores of all algorithms on the Lytro dataset, where the best and the second-best values are highlighted by the red and blue respectively.**

| Method | Lytro Dataset | | | | | | | | | |
|---|---|---|---|---|---|---|---|---|---|---|
| | EN↑ | MI↑ | SF↑ | AG↑ | SD↑ | CC↑ | VIF↑ | $Q_{abf}$↑ | PSNR↑ | SSIM↑ |
| SFMD | 7.5623 | 5.8522 | 19.3574 | 6.0618 | 59.0174 | 0.9598 | 1.0318 | 0.6500 | 72.7596 | 1.3131 |
| DCT_Corr | 7.5330 | 8.4953 | 19.3452 | 6.8160 | 57.4378 | 0.9712 | 1.3526 | 0.7501 | 74.5680 | 1.3554 |
| MGDN | 7.5281 | 6.8109 | 18.6843 | 6.6195 | 56.8484 | 0.9752 | 1.2237 | 0.7350 | 74.2980 | 1.4104 |
| MFFT | 7.5321 | 8.8159 | 19.4706 | 6.9806 | 57.5509 | 0.9713 | 1.3662 | 0.7527 | 74.5892 | 1.3667 |
| FusionDiff | 7.5859 | 6.5013 | 19.4097 | 6.7953 | 64.6829 | 0.9833 | 1.3076 | 0.7185 | 74.5644 | 1.3570 |
| SwinFusion | 7.5333 | 6.3588 | 19.0595 | 6.7930 | 62.3561 | 0.9766 | 1.1554 | 0.6908 | 72.6754 | 1.3306 |
| ZMFF | 7.5256 | 6.5928 | 18.8764 | 6.7497 | 56.9705 | 0.9699 | 1.1721 | 0.7020 | 74.3756 | 1.3525 |
| MUFusion | 7.4726 | 6.1874 | 11.7441 | 4.4285 | 54.2357 | 0.9722 | 1.0047 | 0.5130 | 74.2022 | 1.3412 |
| Ours | 7.5646 | 8.5046 | 19.4271 | 6.8467 | 64.1423 | 0.9798 | 1.3796 | 0.7602 | 74.7680 | 1.4277 |

interaction leads to inconsistencies in pixel density (SF) between fusion results and source images, as well as loss of edge texture ($Q_{abf}$). Therefore, AFIM is necessary for our network.

**Ret-Attention-Based Spatial Information Extraction Module.** RASEM is utilized to encourage the network to perceive focused regions. We replaced RASEM with AFIM while keeping the

network parameters at the same level, to demonstrate the effectiveness of RASEM. Table 3 shows that replacing RASEM with AFIM leads to a decrease in all IQA metrics, especially SD and PSNR. This is because the network loses the ability to build long-range dependencies between source images and struggles to capture the edge

**Table 2: The average scores of all algorithms on the MFFW dataset, where the best and the second-best values are highlighted by the red and blue respectively.**

| Method | MFFW Dataset | | | | | | | | | |
|--------|------|------|------|------|------|------|------|----------------|--------|--------|
| | EN↑ | MI↑ | SF↑ | AG↑ | SD↑ | CC↑ | VIF↑ | $Q_{abf}$↑ | PSNR↑ | SSIM↑ |
| SFMD | 7.1184 | 4.9109 | 22.6258 | 7.5635 | 54.8823 | 0.9400 | 0.7732 | 0.5496 | 71.1784 | 1.0347 |
| DCT_Corr | 7.1818 | 5.2898 | 22.7697 | 7.6263 | 53.6927 | 0.9383 | 0.9302 | 0.6201 | 71.5740 | 1.0295 |
| MGDN | 7.1728 | 5.8827 | 21.6853 | 7.4742 | 54.5829 | 0.9514 | 1.0301 | 0.6273 | 72.4486 | 1.2574 |
| MFFT | 7.1799 | 6.3114 | 22.5132 | 7.6093 | 55.1139 | 0.9454 | 1.1112 | 0.6941 | 71.9840 | 1.1771 |
| FusionDiff | 7.1889 | 5.7032 | 23.2265 | 7.6935 | 60.8880 | 0.9638 | 1.0720 | 0.7016 | 71.9033 | 1.2314 |
| SwinFusion | 7.1050 | 5.6515 | 20.6891 | 7.3872 | 54.1665 | 0.9427 | 1.0076 | 0.6779 | 71.2389 | 1.2300 |
| ZMFF | 7.1711 | 5.5198 | 21.4803 | 7.5699 | 53.9978 | 0.9422 | 1.0011 | 0.6671 | 71.8072 | 1.1465 |
| MUFusion | 7.1675 | 5.4347 | 20.8136 | 7.1053 | 53.6772 | 0.9546 | 0.8934 | 0.5975 | 72.9840 | 1.0844 |
| Ours | 7.1829 | 6.2295 | 22.8462 | 7.6274 | 58.3208 | 0.9598 | 1.1232 | 0.7043 | 73.2130 | 1.2605 |

Near-focused    Far-focused    SFMD    DCT_Corr    MGDN    MFFT    FusionDiff    SwinFusion    ZMFF    MUFusion    Ours

**Figure 6: The visual comparisons between other MFIF methods and our method on the MFFW dataset.**

**Table 3: Ablation studies about the AFIM and RASEM on the Lytro dataset. The best values are bolded.**

| AFIM | RASEM | EN↑ | MI↑ | SF↑ | AG↑ | SD↑ | CC↑ | VIF↑ | $Q_{abf}$↑ | PSNR↑ | SSIM↑ |
|------|-------|-----|-----|-----|-----|-----|-----|------|-----------|-------|-------|
| × | ✓ | 7.5302 | 7.3599 | 18.9873 | 6.7036 | 56.6362 | 0.9716 | 1.3150 | 0.7400 | 74.5153 | 1.3488 |
| ✓ | × | 7.5311 | 7.7320 | 19.3146 | 6.8028 | 57.1946 | 0.9710 | 1.3420 | 0.7465 | 74.4739 | 1.3548 |
| Ours | | **7.5402** | **7.7813** | **19.3924** | **6.8485** | **57.7638** | **0.9720** | **1.3451** | **0.7489** | **74.8243** | **1.3621** |

**Table 4: Ablation studies about the IDFM on the Lytro dataset. The best values are bolded. 'w/o' denotes without, 'w/' denotes with.**

| Config | EN↑ | MI↑ | SF↑ | AG↑ | SD↑ | CC↑ | VIF↑ | $Q_{abf}$↑ | PSNR↑ | SSIM↑ |
|--------|-----|-----|-----|-----|-----|-----|------|-----------|-------|-------|
| w/o IDFM | 7.1149 | 4.6708 | 13.0093 | 5.0832 | 39.5298 | 0.9527 | 0.7954 | 0.5023 | 62.8179 | 0.9567 |
| w/ IDFM | **7.5402** | **7.7813** | **19.3924** | **6.8485** | **57.7638** | **0.9720** | **1.3451** | **0.7489** | **74.8243** | **1.3621** |

**Table 5: Ablation studies about the 2D Ret-Attention on the Lytro dataset. The best values are bolded. 'w/o' denotes without, 'w/' denotes with.**

| Config | EN↑ | MI↑ | SF↑ | AG↑ | SD↑ | CC↑ | VIF↑ | $Q_{abf}$↑ | PSNR↑ | SSIM↑ |
|--------|-----|-----|-----|-----|-----|-----|------|-----------|-------|-------|
| w/o 2D Ret-Attention | 7.5316 | 7.6888 | 19.3551 | 6.8165 | 57.4708 | 0.9712 | 1.3348 | 0.7446 | 74.6069 | 1.3514 |
| w/ 2D Ret-Attention | **7.5402** | **7.7813** | **19.3924** | **6.8485** | **57.7638** | **0.9720** | **1.3451** | **0.7489** | **74.8243** | **1.3621** |

**Table 6: Ablation studies of the loss function terms on the Lytro dataset. The best values are bolded. 'w/o' denotes without.**

| Config | EN↑ | MI↑ | SF↑ | AG↑ | SD↑ | CC↑ | VIF↑ | $Q_{abf}$↑ | PSNR↑ | SSIM↑ |
|--------|-----|-----|-----|-----|-----|-----|------|-----------|-------|-------|
| w/o $\mathcal{L}_{pix}$ | 7.5320 | 7.1007 | 19.3192 | 6.8017 | 57.5521 | 0.9715 | 1.2851 | 0.7467 | 74.6106 | 1.3423 |
| w/o $\mathcal{L}_{grad}$ | 7.5374 | 6.9961 | 19.2612 | 6.8011 | 57.2593 | 0.9711 | 1.2998 | 0.7339 | 74.5454 | 1.3489 |
| w/o $\mathcal{L}_{ssim}$ | 7.5250 | 6.8628 | 19.1148 | 6.7323 | 56.4338 | 0.9711 | 1.3017 | 0.7373 | 74.6134 | 1.3510 |
| Ours | **7.5402** | **7.7813** | **19.3924** | **6.8485** | **57.7638** | **0.9720** | **1.3451** | **0.7489** | **74.8243** | **1.3621** |

detail information within the focused regions, resulting in more noise in the fusion results. Thus, RASEM is crucial in the SFIMFN.

**Invertible Dual-domain Feature Fusion Module.** IDFM is utilized to avoid information loss during the dual-domain information fusion process. To validate its effectiveness, we replaced it with a densely-connected architecture. For fair comparison, we keep the above two comparisons with the same number of parameters. The results in Table 4 demonstrate that removing IDFM significantly weaken our network's performance, highlighting the importance of IDFM in our network.

**2D Ret-Attention.** To further validate the effectiveness of the 2D Ret-Attention mechanism, we replaced it with the Shifted windows attention mechanism from the SwinTransformer [26]. The results in Table 5 demonstrate that the 2D Ret-Attention mechanism significantly improves the performance of the model. Specifically, there is an increase of 0.21 dB in PSNR. Thus, the 2D Ret-Attention plays a crucial role in MFIF.

**Loss Function.** We verified the effectiveness of each loss function by removing them individually, where the results are reported in Table 6. The pixel intensity loss $\mathcal{L}_{pix}$ is employed to reduce the chromatic aberration between the source and fused images, removing $\mathcal{L}_{pix}$ leads to a notable decrease in all metrics. The gradient loss $\mathcal{L}_{grad}$ constrains the fused image to have the same texture detail as

the sharp source images. Therefore, removing $\mathcal{L}_{grad}$ leads to a significant decrease in PSNR and SSIM, 0.27 dB and 0.01, respectively. The structural similarity loss $\mathcal{L}_{ssim}$ constrains the fusion network to maintain the structural information in the source images. In addition, $\mathcal{L}_{ssim}$ could restrain the brightness of the fusion results to some extent. Similarly, the incorporation of the SSIM loss leads to improvements in all metrics, with PSNR and SSIM increasing by 0.21 dB and 0.01, respectively. Consequently, each loss term proves to be effective.

## 5 CONCLUSION

In this paper, we propose a novel unsupervised MFIF network named SFIMFN that efficiently perceives details of focused regions by integrating spatial-frequency dual-domain information. To the best of our knowledge, this is the first attempt to investigate the MFIF task from the perspective of spatial-frequency information integration. Moreover, we design a customized transformer by redesigning the self-attention mechanism into a bidirectional, two-dimensional form of explicit decay to encourage the network to perceive the focused regions more efficiently. Extensive experiments on different datasets demonstrate that our proposed method outperforms existing unsupervised methods in both quantitative and qualitative metrics as well as the generalization ability.

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
