# OpenReview forum: "Efficient Perceiving Local Details via Adaptive Spatial-Frequency Information Integration for Multi-focus Image Fusion"
_acmmm.org/ACMMM/2024/Conference — MM2024 Poster_

### Official Review · Reviewer_sAQG · 2024-05-17

**Rating:** 3
**Confidence:** 4

**Summary:**

In this paper, the authors proposed a novel unsupervised  spatial-frequency interaction multifocus image fusion network named SFIMFN. In the framework, there are three key modules: Adaptive Frequency Domain Information Interaction  Module(AFIM), Ret-Attention-Based Spatial Information Extraction Module (RASEM), and Invertible Dual-domain Feature  Fusion Module (IDFM). Overall, the idea is novel. The paper is well written. However, downstream tasks of MFIF should be compared in the experiments.

**Strengths:**

The authors combine the merits of frequency domain and spatial domain to construct a new MFIF algorithm.
The paper is well written.

**Limitations:**

(1) As we know, the objective of MFIF is to extract the focused information from source images. However, I can not see how you avoid the defocused information in the fused images.
(2) How to determine the balancing factors in the total loss function?
(3) The channel number of featA and featB is?
(4) The superiority of the proposed algorithm is subtle in terms of objective metrics.

**Suitability:**

3

---

### Official Review · Reviewer_QNJ8 · 2024-05-21

**Rating:** 5
**Confidence:** 3

**Summary:**

The research paper introduces an innovative unsupervised methodology designed specifically for multi-focus image fusion. This cutting-edge framework leverages information from both the spatial domain and the frequency domain to achieve superior performance. Through a meticulous integration of these two domains, the proposed approach effectively enhances the quality and clarity of the fused images, ensuring that all relevant details are preserved and highlighted. Comprehensive experimental evaluations demonstrate that this methodology significantly outperforms the current sota unsupervised methods in terms of accuracy, efficiency, and overall image quality. The findings suggest that this novel approach holds great promise for advancing the field of image fusion, offering a robust solution for applications that require precise and reliable image synthesis.

**Strengths:**

1. This paper explores an unsupervised approach dedicated to the MFIF task, which adaptively integrates the spatial-frequency information from multiple source images. The idea is novel, and the methodology described is both intuitive and clear.
2. Comprehensive experiments are conducted to compare the proposed method with a large number of baseline methods. Also, sufficient ablation experiments demonstrate the effectiveness of the proposed modules.
3. I would like to commend the authors on the outstanding quality of the writing in this paper. The clarity and precision with which the ideas are presented significantly enhance the reader's understanding..

**Limitations:**

1. The implementation details of the loss function are not stated clearly enough, the value of the weight factor in Equation (23) is not provided.
2. The explanations for some variables are unclear, including the meanings of v_m and o_n in Equation (9).
3. It seems that the author did not clearly describe how mask_w and mask_h are generated in the method section. Could you provide a more detailed explanation?

**Suitability:**

3

---

### Official Review · Reviewer_QXLy · 2024-05-24

**Rating:** 2
**Confidence:** 3

**Summary:**

This paper introduces a method for multi-focus image fusion.  The authors propose a unsupervised spatial-frequency interaction MFIF network named SFIMFN, which consists of three components: Adaptive Frequency Domain Information Interaction Module, Ret-Attention-Based Spatial Information Extraction Module , and Invertible Dual-domain Feature Fusion Module to integrate spatial-frequency information to achieve MFIF.

**Strengths:**

1.The proposed method is well presented and easily understood.
2.The idea of investigating the MFIF task from the perspective of spatial- frequency information integration is novel.

**Limitations:**

1. The proposed method has no obvious advantages in most metrics. The algorithm's performance is not sufficient for SOTA.
2. The analysis of ablation experiments lacks visual comparison. Authors should provide visual comparisons to further support the effectiveness of their method.

**Suitability:**

2

---

### Official Review · Reviewer_pdzU · 2024-05-28

**Rating:** 4
**Confidence:** 3

**Summary:**

This paper aims to explore the utilization of spatial-frequency information to solve the problem of multi-focus image fusion and proposes a novel unsupervised spatial-frequency interaction multi-focus image fusion network composed of AFIM, RASEM, and IDFM to fully exploit the spatial-frequency information in images.

**Strengths:**

1. The existing problems in multi-focus image fusion are rethought from multiple perspectives, which fully illustrates the rationality of using spatial frequency information.
2. Redesigning the self-attention mechanism into a bidirectional two-dimensional explicit attenuation form allows the network to perceive local focus areas more effectively.
3. The experiment is intuitive and sufficient, and the effectiveness of the proposed network structure is proved through multiple groups of methods and multiple ablation experiments.

**Limitations:**

1. The method section should be concise and concise, highlighting the innovative points of the proposed method as much as possible, and emphasizing important mathematical derivation and formulas.
2. Some indicators such as VIF, SSIM, and PSNR have exceeded the SOTA, but many indicators such as EN and MI have not reached the expected SOTA value. Is it possible to analyze and summarize the reasons?

**Suitability:**

2

---

### Meta-Review · Area_Chair_GAHP · 2024-07-02

**Recommendation:** Accept (Poster)
**Confidence:** 5

**Metareview:**

This paper proposes an unsupervised multi-focus image fusion (MFIF) method, utilizing the spatial-frequency information. The proposed unsupervised spatial-frequency interaction MFIF network consists of three components (AFIM, RASEM, and IDFM) to fully exploit the spatial-frequency information in images. The experimental results demonstrate the effectiveness of the proposed network. It outperforms existing unsupervised fusion methods in terms of various metrics, and also surpasses some supervised fusion approaches with respect to some metrics such as PSNR and SSlM.

After rebuttal, this paper receives 2 positive reviews and 2 negative reviews. Most reviewers agree with the novelty of investigating the MFIF task from the perspective of spatial-frequency information integration, so I recommend the acceptance of this paper.